# Managing the Oral Health of Cancer Patients during the COVID-19 Pandemic: Perspective of a Dental Clinic in a Cancer Center

**DOI:** 10.3390/jcm9103138

**Published:** 2020-09-28

**Authors:** Sunita Manuballa, Marym Abdelmaseh, Nirmala Tasgaonkar, Vladimir Frias, Michael Hess, Heidi Crow, Sebastiano Andreana, Vishal Gupta, Kimberly E. Wooten, Michael R. Markiewicz, Anurag K. Singh, Wesley L. Hicks, Mukund Seshadri

**Affiliations:** 1Department of Oral Oncology/Dentistry and Maxillofacial Prosthetics, Roswell Park Comprehensive Cancer Center, Buffalo, NY 14263, USA; Sunita.Manuballa@RoswellPark.org (S.M.); a.marym1@gmail.com (M.A.); Nirmala.Tasgaonkar@RoswellPark.org (N.T.); Vladimir.Frias@RoswellPark.org (V.F.); Michael.Hess@RoswellPark.org (M.H.); Heidi.Crow@RoswellPark.org (H.C.); Sebastiano.Andreana@RoswellPark.org (S.A.); 2Department of Oral Diagnostic Sciences, University at Buffalo School of Dental Medicine, Buffalo, NY 14214, USA; 3Department of Restorative Dentistry, University at Buffalo School of Dental Medicine, Buffalo, NY 14214, USA; 4Department of Head and Neck/Plastic Surgery, Roswell Park Comprehensive Cancer Center, Buffalo, NY 14263, USA; Vishal.Gupta@RoswellPark.org (V.G.); Kimberly.Wooten@RoswellPark.org (K.E.W.); Michael.Markiewicz@RoswellPark.org (M.R.M.); Wesley.Hicks@RoswellPark.org (W.L.H.J.); 5Department of Oral and Maxillofacial Surgery, University at Buffalo School of Dental Medicine, Buffalo, NY 14214, USA; 6Department of Radiation Medicine, Roswell Park Comprehensive Cancer Center, Buffalo, NY 14263, USA; Anurag.Singh@RoswellPark.org; 7Center for Oral Oncology, Roswell Park Comprehensive Cancer Center, Buffalo, NY 14263, USA

**Keywords:** COVID-19, oral oncology, dental, oral surgery, head and neck cancer, cancer patients

## Abstract

The practice of dentistry has been dramatically altered by the coronavirus disease 2019 (COVID-19) pandemic. Given the close person-to-person contact involved in delivering dental care and treatment procedures that produce aerosols, dental healthcare professionals including dentists, dental assistants and dental hygienists are at high risk of exposure. As a dental clinic in a comprehensive cancer center, we have continued to safely provide medically necessary and urgent/emergent dental care to ensure that patients can adhere to their planned cancer treatment. This was accomplished through timely adaptation of clinical workflows and implementation of practice modification measures in compliance with state, national and federal guidelines to ensure that risk of transmission remained low and the health of both immunocompromised cancer patients and clinical staff remained protected. In this narrative review, we share our experience and measures that were implemented in our clinic to ensure that the oral health needs of cancer patients were met in a timely manner and in a safe environment. Given that the pandemic is still on-going, the impact of our modified oral healthcare delivery model in cancer patients warrants continued monitoring and assessment.

## 1. Introduction

The outbreak of coronavirus disease 2019 (COVID-19) [1] has dramatically changed the practice of dentistry worldwide, given the close “person-to-person” contact involved in delivering dental care and treatment procedures that produce aerosols, often resulting in dental providers being exposed to blood, saliva and respiratory droplets [2,3]. Given the mode of transmission of COVID-19, dental healthcare professionals including dentists, dental assistants and dental hygienists are at high risk of exposure among all healthcare personnel [4,5]. In this regard, studies from multiple groups around the globe have reported on critical infection control measures [5,6,7] and guidelines for modifications to dental clinic workflows that were implemented during the pandemic [8,9]. Reports have also described the impact of COVID-19 on the practice of dental specialties, including oral medicine [10] oral and maxillofacial surgery [11,12,13], orthodontics [14] and endodontics [4,15].

Oral oncology (sometimes referred to as “Dental Oncology”) is a branch of dentistry/oral medicine that provides specialized care to address the complex dental and oral health needs of cancer patients [16,17]. The division of Dentistry and Maxillofacial Prosthetics (DMFP) is a clinical service within the Department of Oral Oncology at Roswell Park, a National Cancer Institute (NCI)-designated Comprehensive Cancer Center located in Buffalo, New York. The center provides comprehensive cancer care to patients in the Buffalo–Niagara metropolitan area, surrounding counties in Western New York (WNY) and patients from New York State (NYS). The center also provides cancer care for patients from other states within the U.S. and Canada. The mission of DMFP is to provide high-quality oral healthcare to cancer patients. Specialized services provided by DMFP include management of existing dental conditions prior to the start of cancer therapy, prevention and management of oral complications from cancer treatment (radiation, chemotherapy and hematopoietic stem cell transplantation) and functional rehabilitation of patients after invasive cancer surgery [18,19,20]. Cancer patients are immunocompromised and, as a result, susceptible to oral and respiratory infections, including COVID-19 [21]. Given this “double whammy” (increased risk for cancer patients and dental providers), the pandemic has necessitated rapid implementation of changes to our oral healthcare delivery model, including adaptation of clinical workflows and diagnostic and treatment paradigms. Kochhar et al. have recently described recommendations for provision of dental care to cancer patients during the pandemic [22]. As a dental clinic in a comprehensive cancer center, we have continued to safely deliver dental care to cancer patients during this pandemic. This was accomplished through adaptation of clinical workflows to ensure that cancer patients can adhere to their planned cancer treatment. Timely implementation of practice modification measures was critical to ensure that patients and dental clinic staff remained protected and the risk of transmission remained minimal.

## 2. Overview of DMFP Clinic Responsiveness to COVID-19

The overview of the DMFP clinic response to COVID-19 is shown schematically in Figure 1. In response to the pandemic, a DMFP clinic task force was created in early March 2020 to implement clinic-centric measures that were in compliance with Centers for Disease Control and Prevention (CDC), the American Dental Association (ADA), NYS and institutional guidelines and develop protocols for safely providing dental care for cancer patients. The taskforce included the department chair, the clinical chief, a general dentist, the lead dental assistant and the clinic administrator. Such a composition of the task force ensured that all administrative and operational needs of the clinic and staff concerns were addressed. Given the relatively fluid nature of the situation, daily virtual meetings of the taskforce were conducted (via WebEx) to monitor the regional situation and to appraise team members of any updates to institutional policies regarding patient care and staff. The goals of the task force and the strategic approach undertaken to implement clinic-centric measures that complemented institutional measures are summarized in Figure 1.

The following measures were implemented in our clinic to limit traffic in clinical areas and ensure that the oral health needs of cancer patients were met in a timely manner and in a safe environment.

### 2.1. Identifying Critical Services Provided by the Dental Clinic

With the evolution of the pandemic in NYS and around the United States, the ADA, CDC and the New York State Dental Association (NYSDA) issued guidance on March 16, 2020, that all dental offices provide only emergency dental care for patients. A minimum of 3 weeks of postponement was recommended for all elective and non-emergent services. By mid-March 2020, the dental clinic taskforce had decided to scale down clinical operations to essential critical functions. In compliance with NYS, ADA and CDC guidelines, clinic visits were restricted to management of active cancer patients that needed medically necessary oral health evaluations (e.g., patients requiring dental clearance prior to start of radiation therapy, bone marrow transplant patients, patients scheduled for surgery) and urgent/emergent dental treatment.

### 2.2. Modifying Clinic Schedules

All clinic providers were asked to review their schedules for the months of March and April. Consistent with the framework [23] suggested by the Centers for Medicare and Medicaid Services (CMS), a three-tiered classification of patients was developed to modify clinic schedules as shown in Figure 2. Patients with scheduled appointments were immediately contacted and their health status assessed over the phone to determine the urgency of their treatment. The immune status of patients (undergoing active chemo/RT or immunotherapy, recent organ transplant; on immunosuppressive therapy) was also taken into consideration while determining the tier and type of appointment. Patients with appointments for elective dental procedures (e.g., routine follow-up appointments, prosthetic or routine oral hygiene maintenance patients that could wait) that could be safely deferred were rescheduled (on an average of 4–6 weeks from the initial appointment date). Patients were notified of their rescheduled appointments by the clinic receptionist along with the communication regarding the availability of all dentists and specialists for telephone consults.

### 2.3. Rotational Scheduling of Clinic Faculty and Staff

In alignment with the institutional “directed to leave campus” (DLC) policy, non-critical clinic staff, including personnel in administrative and research arms of the department, were scheduled to work remotely. Secure remote access (email, virtual desktop) was provided to staff members, which allowed them to continue their daily tasks from home. For essential clinic staff, a rotational schedule was implemented wherein one dentist and two dental assistants were assigned on a weekly basis. The providers and the assistants were on-site two days of the week to attend to patients requiring medically necessary dental procedures, but all providers were available for teleconsultation during the week. Additional emergent/urgent appointments were also handled by the same dentist /dental assistant team scheduled to be “on call” for the week. This arrangement also ensured a two-week window before the same provider/assistant team was scheduled to be back on-site (i.e., Week 1: Team A; Week 2: Team B; Week 3: Team C; Week 4: Team A). This temporal spacing of provider/staff schedules minimized overlap between faculty and staff from individual teams and allowed for a potential 14-day period of isolation or quarantine in the unfortunate event that one of the team members became symptomatic. All staff were instructed to continue self-monitoring and advised to stay home if they experienced flu-like symptoms.

### 2.4. Transitioning to a ‘Virtual’ Tumor Board

Another COVID-related modification in our clinical workflow involved a change in the conduct of a multidisciplinary head and neck conference (“tumor board”). The multidisciplinary conference serves as a valuable forum for discussions among team members regarding diagnosis and treatment planning of head and neck cancer patients. Roswell Park conducts a weekly head and neck tumor board meeting that is attended by faculty from the above-mentioned specialties along with nurse practitioners, physician assistants, physical therapists, palliative care and social workers. These weekly meetings are quintessential to ensure adequate work up for correct diagnosis, staging, discuss treatment strategies (surgery versus radiation), surgical reconstruction approaches for best quality of life and survivorship issues. In a pre-COVID-19 world, this involved an in-person meeting of about 25–30 specialists in a packed conference room. With the onset of the pandemic and the restrictions that followed, such a gathering was no longer possible. Given the large number of participants at these weekly meetings, a decision was made in March 2020 to move the tumor board to an online platform (“virtual” tumor board). We transitioned to weekly virtual tumor board meetings utilizing the WebEx platform developed by Citrix systems. The WebEx platform allowed both video and audio presentation, including a screen sharing ability. The platform was approved by Roswell IT and was compliant with the Health Insurance Portability and Accountability Act (HIPAA) regulations. Email invitations for these sessions were sent out to participants along with the pertinent WebEx information. Participation required a valid attendee name and a Roswell Park email login. The virtual format allowed for efficient participation of a large number of attendees with case presentations made in Microsoft Power Point format by the head and neck fellow, along with review of histology slides by a pathologist and imaging by the radiologist. Although not a new concept, we had no previous experience with virtual multidisciplinary conferences. Our experience to date with the virtual tumor board format has been positive, and the format has encouraged greater and timely participation from a large group of participants. In addition, since most of the attendees participated in the virtual meeting from their workstations, it provided them with the ability to instantly access not just patient records, but also published literature to clarify any points if needed. The biggest limitation of the virtual format is the lack of personal interaction and interactive conversations between multiple speakers. Despite a few initial technical glitches, this approach continues to be effective, allowing the entire team to engage in thoughtful discussions regarding treatment plans for individual patients without increasing the risk of exposure and potential members between clinic staff.

### 2.5. Screening of Patients Prior to Visits

Multiple measures were put in place for screening patients with scheduled appointments at our clinic. These included inquiry of symptoms over the phone (“tele-triage”) 24 h prior to their appointment. Patients were screened (symptom checks, temperature measurements) at the main entrance to our cancer center and then at the clinic front desk at check-in. The duration of appointments for patients requiring medically necessary or emergency dental procedures was also lengthened (approximately 90 min) with adequate time (30–45 min) between appointments to allow clinic staff to perform all infection control procedures according to CDC guidelines.

### 2.6. Infection Control Training and Operatory Preparation

Recognizing the need for stringent infection control protocols, several institutional and clinic-centric training measures were implemented to train all clinical faculty and staff. In addition to the mandatory annual in-services routinely completed by all hospital staff, refresher training on infection control procedures was provided in the form of training videos, instruction sheets, flyers as well as presentations and group discussions (via WebEx). Topics covered in these training sessions included but were not limited to the basics of COVID-19, hand hygiene, respiratory etiquette, personal protective equipment (PPE), donning and doffing, sterilization and disinfection procedures, protection of all equipment (e.g., computer screens, monitors), proper disposal of single-use instruments and minimizing clutter (e.g., leaving all paperwork outside the operatory; removal of any potential sources of contamination, such as pictures from the walls) within the operatories. All providers, including assistants, wore full surgical garb (shoe covers, head covers, surgical gowns, gloves, N95 masks and face shields) while examining patients.

### 2.7. Clinical Care Guidelines for Dental Management of Cancer Patients

Given the unprecedented nature of events, guidelines for managing oral care of cancer patients were developed. Due to the known risk of COVID-19 transmission via respiratory droplets, a decision was made to avoid high aerosol-generating procedures including the use of high-speed hand pieces, ultrasonic scalers and air-water syringe. A prophylactic hydrogen peroxide mouth rinse was provided to all patients prior to their clinical examination. Aerosol-generating procedures were avoided in severely immunocompromised patients, and clinical evaluation of these patients was performed in dedicated operatories. Consideration was given and plans put in place for performing low aerosol-generating procedures in the operating room (OR) to these patients if clinically warranted. Conservative treatment for asymptomatic carious restorations and minimal debridement were provided without the use of ultrasonic equipment to reduce the microbial load prior to their treatments. Restorative procedures were performed following the principles of atraumatic restorative therapy [24]. Based on the depth of invasion and the presence/absence of symptoms, caries excavation and temporary restorations were placed. Alternatively, caries arresting measures, through the use of silver diamine fluoride (SDF 38% Advantage Arrest), were performed. Hand scaling was performed, and no ultrasonic scaling or polishing was done. Procedures were performed under rubber dam isolation, and radiography was limited to extra-oral radiographs. Extractions of non-restorable teeth were performed following the recommended PPE protocols. Extractions of teeth were only done when the teeth had significant mobility, poor bone support or a root morphology that was amenable to a simple extraction. For head and neck patients undergoing surgical procedures, dental extractions were performed in the operating room (OR) in close coordination with the head and neck surgeons. Contingency plans were also made for providing dental care in peri-operative surgical suites as alternatives to dental operatories, if needed. Emergency floor consultations were provided for in-patients and treatments provided as needed. During this time, the dental faculty maintained crucial communication with patients, their treating physicians (medical and radiation oncologists) and surgeons to maintain continuity of care.

### 2.8. Maxillofacial Rehabilitation of Cancer Patients during COVID-19

Maxillofacial prosthetics is the sub-specialty of prosthodontics that deals with the rehabilitation of head and neck defects beyond the immediate oral region. The most common procedures performed by a maxillofacial prosthodontist at a cancer center are the obturation or restoration of missing maxillary and mandibular structures and the replacement of missing orbital, nasal, auricular and cranial structures [16,17,25]. Other procedures performed include surgical placement of implants to support or retain prostheses or the creation of devices to aid the delivery of surgical or radiation treatment. The decision-making process during COVID-19 was complicated by the fact that, although many non-emergent prosthetic procedures could be postponed, the delay in adequate rehabilitation has major consequences, including deficient speech, swallowing as well as the psychosocial issues for patients due to a visibly missing body part. The treatment protocol at Roswell Park considered both the patient’s medical status as well as the urgency of the procedure involved and attempted to provide treatment with the fewest number of visits and procedures to prevent exposure to the virus. Medically necessary surgical obturations that could be inserted with sutures or ligation were performed in the operating room. Removal of the surgical prosthetic and replacement with an interim prosthesis were evaluated on a case-by-case basis, and all adjustments were carried out under a laboratory hood. Since the creation of definitive prosthetics often requires multiple aerosol-generating procedures, these treatments were postponed. Conventional prosthodontic procedures and surgical implant placement also generate a large amount of aerosol and were delayed in order to limit the exposure of severely medically compromised patients. Fabrication of facial or somatic prostheses also requires multiple visits and would increase patient and provider risk of exposure (asymptomatic carriers) and was postponed.

### 2.9. Telemonitoring and Follow-Up

All phone consultations were documented in the electronic health record (EHR). The temporal scheduling of clinical faculty and staff enabled providers that were off-site to monitor requests for clinic appointments. Concerns and requests from patients seeking emergency appointments were reviewed by a dental provider to understand the nature of their emergency and the appropriate course of management. Phone consults were also performed for patients receiving radiation to follow up for dysphagia, mucositis or candidiasis. Individual prescription requests (e.g., fluoride toothpaste, chlorhexidine) were managed by the providers. Appropriate prescriptions were called into their pharmacies. All patient concerns were initially addressed with a phone call, and if a clinic visit was deemed necessary, the patient was scheduled for a visit.

### 2.10. Modifications to a General Practice Dental Residency Training Program

The DMFP department serves as a home to a 1-year General Practice Residency (GPR) program that is administered jointly with the State University of New York, University at Buffalo, School of Dental Medicine. The program has an annual intake of two residents who spend approximately 70 percent of their time in the clinical care of patients. The didactic portion consists of treatment-planning seminars and literature reviews, as well as lectures in diagnosis, prosthodontics, endodontics and practice management. When the directive to leave campus (DLC) for all non-essential staff was implemented at our cancer center, a distance education model based on an online learning curriculum and continuing education (CE) credits covering all areas of general dentistry was implemented. Subsequently, following the implementation of several risk reduction measures and enhanced infection controls, residents were allowed to return on a rotating basis and observe patient care, to minimize contact and exposure for an already vulnerable patient population. PPE donning/doffing procedures were extensively reviewed. Modifications to dental practice and clinical decision making in the context of a pandemic were thoroughly explained. Differences in risk–benefit considerations between ideal treatment plans versus minimally acceptable treatment to “clear” patients for oncologic care were discussed. Residents also participated in care by conducting assessment phone calls with patients calling the clinic with dental concerns. Residents learned to triage and classify dental emergencies and urgent care cases based on ADA interim guidelines. Emphasis was placed on gathering pertinent patient information to come up with working diagnoses before scheduling patients to minimize appointment times and overall exposure. Residents were tasked with leading virtual weekly case reviews. A thorough medical and dental history was presented for each patient that was seen in the clinic that week. Dental treatment that was proposed or completed to clear them to proceed with their cancer care was outlined and discussed with the faculty. Detailed discussions took place surrounding the modifications that had to be made in their oncologic and dental care as a result of COVID-19. Although not ideal, these modifications enabled the residents to effectively continue their dental and oral oncology training during the pandemic.

## 3. Conclusions

### 3.1. Teamwork Is Integral to Ensure Timely and Optimal Coordination of Care

Managing the oral health of cancer patients requires timely coordination of care between surgeons, radiation oncologists, medical oncologists, radiologists, pathologists, dentists and maxillofacial prosthodontists. For example, we have previously documented that stem cell transplant patients may have a narrow window of adequate disease control for successful transplantation [26]. Similarly, the time from a head and neck cancer diagnosis to the initiation of radiation therapy correlates with survival [27]. As a result, timely dental clearance of cancer patients is integral to their overall cancer care. While such coordination of care was routinely performed prior to the pandemic, the unprecedented outbreak of COVID-19 posed several logistical challenges and uncertainties in dental and oral healthcare delivery models. Therefore, the importance and the value of teamwork and communication between dentists and medical professionals cannot be understated.

### 3.2. The Road Ahead

We have recently begun resuming our clinical services in a phased manner with modified clinical workflow while maintaining social distancing guidelines in our clinic waiting rooms and continued incorporation of infection control procedures in our clinic areas. At the present time, clinical care is provided with staggered appointments while simultaneously managing unscheduled consultation requests and emergencies. Roswell Park currently offers COVID testing on-site by scheduled appointment, drive-up or in an expedited manner at point-of-care for cancer patients based on clinical need. As a result, we have routinely begun COVID testing our dental clinic patients. These modifications have enabled us to safely provide dental care to cancer patients while ensuring that they adhere to their planned cancer treatment. However, it is now being increasingly recognized that the impact of COVID-19 is likely to be long-standing (several months to years) with a possibility of a second wave of infections in the fall. As we gradually progress to a “new normal”, evaluating the impact of these optimized practices and processes within the dental clinic will be important. Given the planned scale-down of our clinic operations, the number of patients seen by our clinic during this time was reduced. We are currently reviewing our clinic volume data (number of appointments, number and type of procedures performed) during March–September 2020 and comparing it to our “pre-COVID” (March–September 2019) metrics. Such a comparative assessment would have to take additional variables into consideration, including number of active providers (dentists and specialists) on staff, number of residents and so on., to recognize the true impact of COVID-19 on our practice. Equally important is measuring the impact of these practice modifications on patient outcomes and evaluating the overall impact of these changes to clinical workflows on patient experience. The impact of COVID-19 on healthcare economics for dentistry and oral medicine also warrants further investigation. In this regard, we have recently begun examining the financial consequences of COVID-19 on our clinical practice through a review of our billing records. We continue to closely monitor the impact of our clinic measures on treatment-related oral health complications and outcomes in cancer patients and hope to report our findings in the future.

## Figures and Tables

**Figure 1 jcm-09-03138-f001:**
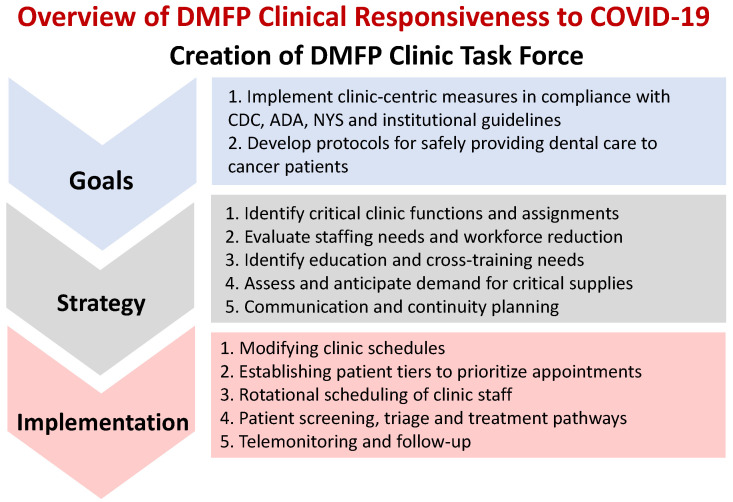
Schematic overview of the responsiveness of the Dentistry and Maxillofacial Prosthetics (DMFP) clinic at Roswell Park Comprehensive Center to the pandemic. (CDC—Centers for Disease Control and Prevention; ADA—American Dental Association; NYS—New York State).

**Figure 2 jcm-09-03138-f002:**
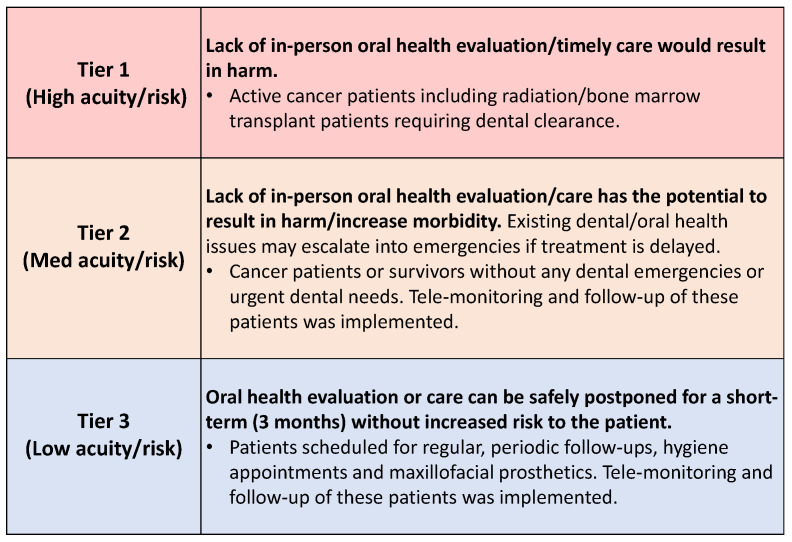
Three-tiered classification of patients based on their oral health needs for optimal scheduling of appointments during the pandemic.

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
