# Peer review of "Managing the Oral Health of Cancer Patients During the COVID-19 Pandemic: Perspective of a Dental Clinic in a Cancer Center"

_jcm, 2020, doi:10.3390/jcm9103138_

Round 1
Reviewer 1 Report
Please consider citing in the introduction among the reports that have described the impact of COVID-19 on the practice of dental specialties a recent study published in this journal that describes the impact of covid 19 on orthodontics (PMID: 32560322)
Please clarify the design of your study, e.g. narrative review or else
Author Response
Reviewer 1
- Please consider citing in the introduction among the reports that have described the impact of COVID-19 on the practice of dental specialties a recent study published in this journal that describes the impact of COVID-19 on orthodontics (PMID: 32560322).
Thank you. We have now included this citation in the introduction (ref#14)
- Please clarify the design of your study, e.g. narrative review or else.
This has now included in the abstract (line 27).
Reviewer 2 Report
I have reviewed manuscript entitled "Managing the Oral Health of Cancer Patients. During 3 the COVID-19 Pandemic: Perspective of a Dental 4 Clinic in a Cancer Center" .This is a manuscript describing management and protocols of dental treatment during covid-19 pandemic.
Overall it is interesting to see the adjustments that were made in the department and especially in the context of medically compromised patients (cancer patients) during COVID-19 pandemic.
Few comments:
1/ Most important, the manuscript contain many details, protocols and modifications in comparison to regular times. You have to had tables that will bring this information in a more arranged way. Also a flow chart will be a very valuable addition
2/ Was there a difference in treatment strategy while dealing with severe cancer patients or immunocompromised patients compared to for easy (less ill) patients?
3/ Do you have any numbers of treatments / patients during this time compared to last year? Did the numbers of treatments dropped?
4/ How do you manage patients concerns?
5/ The manuscript is a written an a bit confused way, please re-organize
Author Response
Responses to Critiques Manuscript ID: JCM-913625
Reviewer 2
I have reviewed manuscript entitled "Managing the Oral Health of Cancer Patients. During 3 the COVID-19 Pandemic: Perspective of a Dental 4 Clinic in a Cancer Center". This is a manuscript describing management and protocols of dental treatment during Covid-19 pandemic. Overall, it is interesting to see the adjustments that were made in the department and especially in the context of medically compromised patients (cancer patients) during COVID-19 pandemic.
We appreciate the reviewer’s comment and thoughtful review of our manuscript.
The manuscript contains many details, protocols and modifications in comparison to regular times. You have to add tables that will bring this information in a more arranged way. Also a flow chart will be a very valuable addition.
We thank the reviewer for this valuable suggestion. We have now included several schematics and tables to illustrate modifications to our workflow.
- Was there a difference in treatment strategy while dealing with severe cancer patients or immunocompromised patients compared to for easy (less ill) patients?
Indeed, the immune status of the patient was considered while scheduling appointments and developing a safely implementable dental treatment plan for patients. Aerosol-generating procedures were avoided in severely immunocompromised patients and clinic visits of these patients was performed in dedicated operatories. Consideration was given and plans put in place for performing low- aerosol generating procedures in the operating room (OR) in these patients if clinically warranted. Conservative treatment was provided for asymptomatic carious restorations and minimal debridement were provided without use of ultrasonic equipment to reduce the microbial load prior to their treatments. Extractions of non-restorable teeth were performed following recommended PPE protocols. We have now included the tiered classification of patients that was employed to modify clinic schedules and safely postpone patient appointments.
- Do you have any numbers of treatments / patients during this time compared to last year? Did the numbers of treatments dropped?
Indeed, given the planned scaling down of our clinic operations by rescheduling of non-emergent/urgent clinic appointments, the number of patients seen by our clinic during this time was significantly reduced. We are currently evaluating and analyze our clinic volume data (number of appointments, number and type of procedures performed) during March-September 2020 comparing it to our pre-COVID (Mar-Sept 2019 metrics). Such a comparative assessment would need to account for additional variables including number of active providers (dentists and specialists) on staff, number of residents etc. to fully recognize the impact of COVID on our practice. We plan on reporting the results of our analysis in the future. This has now been stated in the revised manuscript.
- How do you manage patients concerns?
Patient concerns and requests from patients seeking emergency appointments were reviewed by a dental provider to understand the nature of their emergency and the appropriate course of management. Individual prescription requests (e.g. fluoride toothpaste, chlorhexidine) were managed by the providers. Appropriate prescriptions were called into their pharmacies. All patient concerns were initially addressed with a phone call and if a clinic visit was deemed necessary, the patient was scheduled for a visit. This has now been clarified in the revised version.
- The manuscript is a written in a bit confused way, please re-organize
We have now included re-organized the manuscript and included schematics and illustrations for improved clarity.
Reviewer 3 Report
The Authors proposed an interesting paper focusing on the managing the Oral Health of Cancer Patients During the COVID-19 Pandemic.
The paper is well written and uptodate.
The Authors proposed a valid point of view regard the management of oral cancer during the pandemic.
The interest for the Journal readers is high.
No major issues should be addressed. I just suggest to improve bibliography with adequate references regarding the impact of covid-19 epidemic on maxillofacial surgery in the world.
Author Response
Responses to Critiques Manuscript ID: JCM-913625
Reviewer 3
The Authors proposed an interesting paper focusing on the managing the Oral Health of Cancer Patients During the COVID-19 Pandemic. The paper is well written and uptodate. The Authors proposed a valid point of view regard the management of oral cancer during the pandemic. The interest for the Journal readers is high. No major issues should be addressed. I just suggest to improve bibliography with adequate references regarding the impact of covid-19 epidemic on maxillofacial surgery in the world.
We appreciate the reviewer’s comments. We have updated the bibliography by including additional references in the context of maxillofacial surgery (refs.11-13).
Reviewer 4 Report
I read with great interest this manuscript, which surely addresses a timely issue.
I would suggest the authors to add data on the number of patients treated, and if any cases of COVID-19 occurred among your patients and/or your staff.
Author Response
Responses to Critiques Manuscript ID: JCM-913625
Reviewer 4
I read with great interest this manuscript, which surely addresses a timely issue. I would suggest the authors to add data on the number of patients treated, and if any cases of COVID-19 occurred among your patients and/or your staff.
We are currently reviewing our clinic volume data (number of appointments, number and type of procedures performed) during March-September 2020 comparing it to our pre-COVID (Mar-Sept 2019 metrics). Such a comparative assessment would need to account for additional variables including number of active providers (dentists and specialists) on staff, number of residents etc. to fully recognize the impact of COVID on our practice. We plan on reporting the results of our analysis in the future. This has now been stated in the revised manuscript.
Reviewer 5 Report
"Managing the Oral Health of Cancer Patients During 2 the COVID-19 Pandemic: Perspective of a Dental 3 Clinic in a Cancer Center” is a study detailing experiences and countermeasures at the Dental Clinic in a Cancer Center against COVID-19. This clinical paper is very interesting. However, there are corrections that are essential to meet the standard for publication. Please refer to the following comments.
- The authors have described measures to provide safe dental treatment within the many restrictions of the COVID-19.
However, its effect remains unknown. The interest of dentists is the effect of COVID-19 measures.
How has the exacerbation of symptoms in dental patients increased? Has the economics of dentistry changed?
In other countries, dentists are taking action in response to outbreaks of COVID-19.
Please indicate the situation of COVID-19 and the effect of countermeasures accordingly.
- Please compare and explain the countermeasures that your facility has paid particular attention to in comparison with the various countermeasures for COVID-19 in the world.
Author Response
Responses to Critiques Manuscript ID: JCM-913625
Reviewer 5
Managing the Oral Health of Cancer Patients During 2 the COVID-19 Pandemic: Perspective of a Dental 3 Clinic in a Cancer Center” is a study detailing experiences and countermeasures at the Dental Clinic in a Cancer Center against COVID-19. This clinical paper is very interesting. However, there are corrections that are essential to meet the standard for publication. Please refer to the following comments.
We appreciate the thoughtful feedback provided by the reviewer.
The authors have described measures to provide safe dental treatment within the many restrictions of the COVID-19. However, its effect remains unknown. The interest of dentists is the effect of COVID-19 measures. How has the exacerbation of symptoms in dental patients increased?
The reviewer’s raises an important point. To this end, we are currently conducting review of electronic health records of our patients to document patient outcomes including worsening of symptoms (if any) during March-June 2020 (peak of COVID in NY). We plan on comparing it data from our pre-COVID (Mar-Sept 2019 metrics). We plan on reporting our findings in the near future. This has now been stated in the revised manuscript (Discussion).
Has the economics of dentistry changed?
The reviewer’s raises another important question regarding the impact of COVID on healthcare economics for dentistry/oral medicine. However, we have only recently begun examining the financial consequences of COVID on our clinical practice and cannot provide specific information at this point. As stated above, we plan on evaluating the financial impact of COVID through electronic review of health records and our billing records. We have now included additional lines recognizing the issue raised by the reviewer in the Discussion section of the revised manuscript.
In other countries, dentists are taking action in response to outbreaks of COVID-19. Please indicate the situation of COVID-19 and the effect of countermeasures accordingly. Please compare and explain the countermeasures that your facility has paid attention to in comparison with the various countermeasures for COVID-19 in the world.
We have now included figures to summarize our responsiveness and the tiered classification of patients that was employed to modify clinic schedules and safely postpone patient appointments.
Our measures taken were per the guidelines of the American Dental Association, the New York State Dental Society, the American Association of Oral and Maxillofacial Surgeons, and the New York State Society of Oral and Maxillofacial Surgeons. These clinic-centric measures were also consistent with the guidelines of the European, Asian, and Dental and Oral and Maxillofacial Societies around the world.
Reviewer 6 Report
1. It seems important to clarify what cases fall into the category "urgent" and in what clinical situations (in Covid-19 pandemic) it is recommended to postpone the oncologic' patient's appointment?
2. I deal mostly with children patients and from my experience, the biggest problem during the pandemic time was canceling dental check-up of oncologic patients due to the fear of virus transmission in the dental operatory room.It has resulted in treatment delays and severe complications at the beginning of the pandemic . Approximately 30% of parents did not show with their children. We found it valuable to implement psychological support for parents/caregivers discussing risks associated with pandemic and necessary dental intervention. Could you share your experience? Do you have statistics on how many patients postponed appointments? Do you have statistics on how many procedures/appointments did you perform March/April 2020 compared to 2019?
Author Response
Responses to Critiques Manuscript ID: JCM-913625
Reviewer 6
- It seems important to clarify what cases fall into the category "urgent" and in what clinical situations (in Covid-19 pandemic) it is recommended to postpone the oncologic' patient's appointment?
As requested by the reviewer, we have now included the tiered classification of patients that was employed to modify clinic schedules and safely postpone patient appointments.
- I deal mostly with children patients and from my experience, the biggest problem during the pandemic time was canceling dental check-up of oncologic patients due to the fear of virus transmission in the dental operatory room. It has resulted in treatment delays and severe complications at the beginning of the pandemic. Approximately 30% of parents did not show with their children. We found it valuable to implement psychological support for parents/caregivers discussing risks associated with pandemic and necessary dental intervention. Could you share your experience? Do you have statistics on how many patients postponed appointments? Do you have statistics on how many procedures/appointments did you perform March/April 2020 compared to 2019?
Given the planned scale down of our clinic operations by rescheduling of non-emergent/urgent clinic appointments, the number of patients seen by our clinic during this time was significantly reduced. We are currently evaluating and analyze our clinic volume data (number of appointments, number and type of procedures performed) during March-September 2020 comparing it to our pre-COVID (Mar-Sept 2019 metrics). Such a comparative assessment would need to account for additional variables including number of active providers (dentists and specialists) on staff, number of residents etc. to fully recognize the impact of COVID on our practice. We are also currently conducting review of electronic health records of our patients to document patient outcomes including worsening of symptoms (if any) during March-June 2020 (peak of COVID in NY). We plan on comparing it data from our pre-COVID (Mar-Sept 2019 metrics). We plan on reporting our findings in the future. This has now been stated in the revised manuscript (Discussion).
Round 2
Reviewer 2 Report
You said in your comments that :"We have now included several schematics and tables to illustrate modifications to our workflow."
I saw no tables only two figures, I think that these must be included
Author Response
We apologize for the confusion.
During the revision, the figures were initially prepared as multiple independent schemas (a schema for workforce goals and strategy, a schema for our adaptations to workflow and a table to describe our patient tiers).
However, for clarity and ease, these multiple schemas were combined to create Figure 1.
Similarly, the table of patient tiers was also modified and presented as Figure 2.
Reviewer 5 Report
Thank you for giving me this opportunity to re-review your revised manuscript.
I am happy that all of the suggested corrections have been made.
Thank you for spending so much effort.
Author Response
We are appreciative of the reviewer's comments. Thank you.